# Math2Sym: A System for Solving Elementary Problems via Large Language Models and Symbolic Solvers

**Minh Phu Nguyen**[1]    **Minh Phuong Pham**[1]    **Tuan Minh Kha**[1,2†]    **Minh Man Ngo**[1,2]

[1]VNUHCM-University of Science
[2]Maverick AI
nguyenminhphu200@gmail.com , pmphuong1704@gmail.com
kha.minh@finsavvy.vn, nmman@hcmus.edu.vn

## Abstract

Traditional models for solving math word problems (MWPs) often struggle to capture both linguistic context and arithmetic reasoning. We propose Math2Sym, a novel approach integrating large language models (LLMs) with symbolic solvers. This method leverages LLMs' language comprehension and symbolic computation's precision to efficiently convert MWPs into solvable symbolic form. We introduce the EMSF dataset for training models to formalize math problems across various complexities. On our defined test set benchmark, fine-tuned models outperform GPT-3.5 by 17% in few-shot tasks and perform comparably to GPT-4-mini on elementary math problems.[1]

## 1 Introduction

Math word problems (MWPs) present a unique challenge in artificial intelligence, requiring the integration of linguistic comprehension and mathematical reasoning to solve questions based on contextual descriptions [1]. The primary obstacles lie in understanding the problem's context and translating it into appropriate mathematical operations, particularly when mapping linguistic information to complex mathematical expressions [2].

MWP solving has evolved from early rule-based systems like STUDENT [3] to machine learning methods [4], improving accuracy but still facing challenges in complex domains. Recent advancements include Chain-of-Thought (CoT) prompting [5], which enhances reasoning by breaking down problems into structured steps, and Program-Aided Language models (PaL) [6], which generate Python code for external computation.

Our work advances the field with the following key contributions:

- We introduce a novel approach that fine-tunes models to generate symbolic forms of MWPs, enhancing language models' capabilities in converting problems into representations compatible with our custom SymPy-based solver [7].

- We present the EMSF dataset for converting elementary math word problems into symbolic form across five problem types, facilitating improved formalization of MWPs.

- We demonstrate that fine-tuning 7B-parameter models on EMSF outperforms larger models such as GPT-3.5 in MWP solving.

---

[1]The code and the new EMSF dataset are available at https://github.com/pepoo20/Math2Sym, https://huggingface.co/datasets/MathSymbol/EMSF

[†] Corresponding author

38th Conference on Neural Information Processing Systems (NeurIPS 2024).

This approach enhances MWP solving accuracy and versatility, paving the way for more robust AI systems in mathematical problem-solving. Its transparent step-by-step reasoning also offers educational value, fostering a deeper understanding of problem-solving processes.

## 2 Related Work

### 2.1 MWP Solvers

MWP solving has evolved from early rule-based systems like STUDENT [8], which relied on predefined schemas, to statistical machine learning models [4], improving the mapping from linguistic input to mathematical representations. Deep learning approaches, such as encoder-decoder architectures, further advanced the field [9]. However, many models remain limited to basic arithmetic problems or linear equations, struggling with more complex tasks like systems of equations or inequalities.

### 2.2 Integration of External Tools with Language Models

Recent research has focused on enhancing language models (LMs) by integrating external tools like calculators, search engines, and symbolic solvers to address limitations in precise calculations or accessing real-time information [10, 11]. Two main approaches for training LMs to use these tools have emerged: creating large, supervised datasets with explicit examples of tool usage and using few-shot learning with prompts demonstrating tool use [6, 12].

### 2.3 Auto-Formalization

Auto-formalization, the task of converting natural language into symbolic representations, plays a central role in mathematical reasoning. Recent work in this area leverages symbolic manipulation tools like SymPy [7], alongside proof assistants such as Isabelle/HOL [13], to enable computational formal reasoning. Unlike [14], which uses BERT for simpler problems, and [15], which employs LLMs via prompting without any fine-tuning, our method targets more complex problem types with enhanced LLM-based approaches.

## 3 Math2Sym

Math2Sym integrates large language models (LLMs) with symbolic solvers to address math word problems (MWPs). By transforming natural language problem descriptions into symbolic representations, this approach tackles two key challenges: understanding linguistic complexity and ensuring precise computation. LLMs extract variables and conditions from word problems, while a symbolic solver handles mathematical computations.

### 3.1 Method Framework

Math2Sym converts natural language word problems into standardized *Symbolic Forms* through three core steps:

1. **Extraction of Variables:** Identify relevant entities (variables, constants, relationships) from the natural language description.

2. **Formulation of Mathematical Expressions:** Formalize extracted elements into precise mathematical expressions, adhering to the problem's logic and conditions.

3. **Conversion to Symbolic Form:** Transform mathematical expressions into a predefined *Symbolic Form* compatible with a symbolic solver.

To illustrate this process, consider the following word problem and its symbolic formalization:

**Word Problem:** The length of a rectangle is equal to triple the width. Which system of equations can be used to find the dimensions of the rectangle if the perimeter is 86 centimeters?

**Answer:** Define the variables and formulate the linear system of equations: Let variable $x$ represent the length of the rectangle and variable $y$ represent the width of the rectangle. The length of a rectangle is equal to triple the width, so the equation is $x = 3y$. The perimeter of the rectangle is 86 centimeters, leading to the equation $2x + 2y = 86$.
System of equations: $\{x = 3y, 2x + 2y = 86\}$
Symbolic Form: $[x - 3y, 2x + 2y - 86, x, y, \text{solve}]$

This structured symbolic form provides the solver with a clear and unambiguous mathematical representation, ensuring consistent and accurate solutions across various problem types.

## 3.2 Symbolic Solver

The problem-solving process involves the language model systematically normalizing problems into standard forms, which are then converted into predefined Symbolic Forms. To mitigate the frequent arithmetic errors produced by language models, our approach trains the model to formalize problems while avoiding direct calculations. Unlike other LLMs that attempt computations within their reasoning steps, we delegate all arithmetic to an external solver in Symbolic Form, allowing the model to focus on formalization.

Our solver is built using SymPy [7], a Python library for symbolic computation. SymPy's capability to handle a wide range of mathematical problems and its ease of use make it suitable for both current needs and future scalability. The Symbolic Form follows this structure:

```
Symbolic Form: [[constants or expressions, variables, actions]]
```

Each mathematical problem type is associated with a specific action, which corresponds directly to a SymPy method (e.g., 'solve' for equations or inequalities, 'igcd' for greatest common divisors).

To address LLMs' tendency to produce lengthy outputs, we enclose Symbolic Form answers in double square brackets. This formatting is achieved through prompting or fine-tuning with structured data, facilitating the consistent conversion of problems into Symbolic Forms.

# 4 Experiments

We developed a custom test dataset of 92 questions across five categories: greatest common divisor, least common multiple, systems of equations, linear inequalities, and compound inequalities. Questions vary in difficulty (levels 1-5) and include both word and purely mathematical problems to assess generalization. The questions in the dataset were inspired by problems found in high school mathematics textbooks and reputable online educational resources.

## 4.1 Training Dataset and Models

Language models ranging from a few hundred million to approximately 7 billion parameters were fine-tuned on the EMSF dataset using Low-rank adaptation (LoRA) [16]. Detailed parameters are in A.

The EMSF dataset consists of three parts, as detailed in B:

- Pretrain: Focuses on standard mathematical formalization.

- Basic: Synthetically generated using Mixtral 8x7B [17], involves direct extraction of simple problem elements.

- Advanced: Generated using LLaMA3 70B [18], requires reasoning steps for complex problems.

In short, the basic dataset involves direct extraction and declaration of simple problem elements, whereas the advanced dataset requires reasoning steps and aggregation of information from more complex problems.

## 4.2 Answer Evaluation

Performance was evaluated through direct evaluation, few-shot learning, and two fine-tuned settings (on basic and on advanced datasets). Model-generated symbolic forms were compared to ground truths, both processed through the symbolic solver.

Evaluation scores were weighted by problem's difficulty, with the total score $S_{\text{total}}$ calculated as:

$$S_{\text{total}} = \sum_{i=1}^{N} C_i \times D_i$$

where $C_i \in \{0, 1\}$ is the correctness score for problem $i$, and $D_i \in \{1, 2, 3, 4, 5\}$ is based on problem's difficulty.

## 5 Results

Table 1: Score on our test dataset for the few-shot PaL and solver, both ran on 5-shot prompt with specific prompt for each type of problem

|  | Zero-shot-CoT | Few Shot PaL | Few Shot + Solver | Fine-tuned basic | Fine-tuned advanced |
|---|---|---|---|---|---|
| Mistral 7B | 69 | 113 | 155 | 183 | **210** |
| Mistral 8x7B | 135 | 145 | 154 | Nan | Nan |
| Orca 7B | 25 | 78 | 76 | 133 | **171** |
| Qwen 7B | 116 | 126 | 139 | 170 | **207** |
| WizardMath 7B | 133 | 135 | 140 | 183 | **217** |
| LLama3.1 8B | 156 | 140 | 168 | 197 | **211** |
| Qwen 0.5B | 15 | 22 | 7 | **168** | 132 |
| Qwen 1.8B | 24 | 76 | 52 | **153** | 136 |
| Gemma 2B | 10 | 20 | 39 | 133 | **135** |
| GPT-neo 350M | 0 | 70 | 36 | **130** | 120 |
| Qwen2-Math-Instruct 7B | 163 | **170** | 144 | Nan | Nan |
| GPT 3.5 | 172 | 171 | **179** | Nan | Nan |
| gpt-4o mini | **217** | 182 | 182 | Nan | Nan |

Max score: 231

**Enhanced Performance with Solver Integration**: Our experiments demonstrate that integrating symbolic solvers with LLMs significantly improved performance across models like Mistral 7B, Qwen 7B, WizardMath 7B, and GPT-3.5. This integration outperformed approaches such as Program-aided Language models (PaL) and Zero-shot Chain-of-Thought (CoT) prompting. For instance, Mistral 7B showed a 37% improvement with solver integration compared to Few-Shot PaL, while Qwen 7B demonstrated a 10% increase in performance. These findings underscore the efficacy of the Math2Sym framework, which leverages symbolic solvers to enhance the natural language comprehension and reasoning capabilities of LLMs.

**Comparison with Qwen2-Math**: Our approach outperformed Qwen2-Math-Instruct 7B [19] in overall performance across problem complexities. While Qwen2-Math-Instruct excelled in high-difficulty problems (scoring 170 and solving 5/6 of the most difficult problems), it showed inconsistencies on simpler tasks. In contrast, our models, particularly WizardMath 7B, maintained consistent performance across all difficulty levels, achieving a total score of 217. This consistency demonstrates Math2Sym's versatility in handling both simple and complex tasks.

**Dataset-Driven Success in High-Difficulty Problem Solving** : Models fine-tuned on our EMSF dataset excelled in high-difficulty problems (levels 4 and 5). WizardMath 7B achieved a score of 217, significantly outperforming Zero-shot CoT (133) and GPT-3.5-turbo's best in-context learning (179). This success stems from our dataset's ability to teach diverse problem-to-symbolic-form mapping,

enhancing solver utilization. Notably, our approach yielded results comparable to GPT-4o Mini, demonstrating its competitiveness in challenging problems.

**Performance of Smaller Models in Specific Contexts**: In specific contexts, models with fewer than 1 billion parameters occasionally outperformed mid-sized models. For instance, the fine-tuned Qwen 0.5B model scored approximately 10% higher than the Qwen 1.8B model, suggesting that smaller models may benefit from improved learning efficiency before encountering overfitting issues. However, while smaller models excelled in simpler tasks, larger models like WizardMath 7B consistently outperformed them on complex problems, highlighting the importance of model size in managing problem complexity (See 1).

**Influence of Dataset Complexity on Model Performance**: Our findings reveal that dataset complexity plays a pivotal role in determining model performance. Smaller models excelled on the Basic dataset, but struggled with the Advanced dataset. For instance, Qwen 0.5B's performance dropped by almost 21% when moving from Basic to Advanced tasks. Conversely, larger models like WizardMath 7B improved by about 19% on the Advanced dataset compared to the Basic one. These results highlight the importance of aligning dataset complexity with model capacity, especially for tasks requiring advanced reasoning skills (See 2).

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

# A    Training parameter

For fine-tuning the models in our experiments, we used LLama-Factory [20], an open-source framework for inference and fine-tuning of language models. All fine-tuning was performed on a single NVIDIA L4 GPU using the LoRA method. The hyperparameters used during training are as follows, with all other parameters set to their default values:

- LoRA: rank = 16, alpha = 16, learning rate = 5e-05

The training process consisted of three steps:

1. Pre-training with a dataset containing non-word problems.
2. Supervised fine-tuning (SFT) on the Basic dataset.
3. Supervised fine-tuning (SFT) on the Advanced dataset.

The system prompt used during training was:

> "You are a Math Teacher. Your goal is to understand a math word problem, recognize and distinguish the type of problem, define the variables (if needed), and formulate the problem in symbolic form."

# B    Data

## B.1    Data Leakage

To mitigate data leakage in our dataset, we employed FAISS [21] and Sentence Transformers [22] for vector embedding and similarity filtering. Each mathematical word problem was embedded into a numerical vector using the all-MiniLM-L6-v2 pretrained model from Sentence Transformers.

We then applied FAISS to detect and filter out similar data points, utilizing a cosine similarity threshold of 0.8. Any pairs of word problems exceeding this threshold were removed to prevent overlap between the training and test sets. This approach ensured the integrity of the dataset, minimizing leakage and maintaining valid evaluation metrics for model performance.

## B.2    Pre-train Dataset

The pre-train dataset comprises 1 million samples generated from standard mathematical problems with randomly selected constants and parameters. These problems are converted into symbolic form, providing the model with a foundation for handling structured mathematical tasks without natural language.

Table 2: Examples of pre-train dataset

| Question | Symbolic Form |
|---|---|
| Solve the given system of equations: $x + y = 1, -3x - 6y = 9$ | $[x + y - 1, -3x - 6y - 9, x, y, \text{solve}]$ |
| Inequality: $14x - 7 < 12x + 19$ | $[14x - 7, <, 12x + 19, x, \text{solve}]$ |
| Solve this compound inequality: $5 < -11x + 50 < 6$ | $[5 < -11x + 50, -11x + 50 < 6, x, \text{solve}]$ |
| Find the greatest common divisor of 18 and 27 | $[18, 27, \text{igcd}]$ |
| Calculate the least common multiple of 25 and 92 | $[25, 92, \text{ilcm}]$ |

## B.3    Basic Dataset

The basic supervised fine-tuning (SFT) dataset, containing 149,000 samples, was generated using the Mixtral 8x7b model [17] with few-shot prompting. Data generation parameters were set at

temperature = 1.5, top_p = 0.9, and max_length = 512. The dataset consists of 38% single-variable and compound inequalities, 28% two-variable system of equations, and 17% greatest common divisor and least common multiple problems.

Table 3: Examples of basic dataset

**System of equations**

| Question | Answer |
| --- | --- |
| Word Problem: The length of a rectangle is equal to triple the width. Which system of equations can be used to find the dimensions of the rectangle if the perimeter is 86 centimeters? | Define the variables and formulate the linear system of equations: Let $x$ represent the length of the rectangle and $y$ represent the width of the rectangle. The equation is $x = 3y$. The perimeter is 86 centimeters: $2x + 2y = 86$. System of equations: $\{x = 3 * y, 2 * x + 2 * y = 86\}$ Symbolic form: $[x - 3 * y, 2 * x + 2 * y - 86, x, y, \text{solve}]$ |

**Inequalities**

| Question | Answer |
| --- | --- |
| Word Problem: A taxi charges a flat rate of \$1.75, plus an additional \$0.65 per mile. If Erica has at most \$10 to spend on the cab ride, how far could she travel? | Define the variables and formulate the inequality: Let $d$ represent the distance. The total cost is represented by $1.75 + 0.65 * d <= 10$. Inequality: $1.75 + 0.65 * d <= 10$ Symbolic form: $[1.75 + 0.65 * d <= 10, d, \text{solve}]$ |

**Compound inequalities**

| Question | Answer |
| --- | --- |
| Word Problem: About 20% of the time you sleep is spent in rapid eye movement (REM) sleep. If an adult sleeps 7 to 8 hours, how much time is spent in REM sleep? | Define the variables and formulate the compound inequality: Let $r$ represent time spent in REM sleep. The inequality is $0.2 \times 7 <= r <= 0.2 \times 8$. Compound inequality: $0.2 \times 7 <= r <= 0.2 \times 8$ Symbolic form: $[r - 0.2 \times 7 \geq 0, r - 0.2 \times 8 <= 0, r, \text{solve}]$ |

**Greatest common divisor**

| Question | Answer |
| --- | --- |
| Word Problem: Sara has 16 red flowers and 24 yellow flowers. She wants to make bouquets with the same number of each color flower in each bouquet. What is the greatest number of bouquets she can make? | Find the greatest common factor of 16 and 24. Symbolic form: $[16, 24, \text{igcd}]$ |

**Least common multiple**

| Question | Answer |
| --- | --- |
| Word Problem: Today, both the soccer team and the basketball team had games. The soccer team plays every three days, and the basketball team plays every five days. When will both teams have games on the same day again? | Find the least common multiple of 3 and 5. Symbolic form: $[3, 5, \text{ilcm}]$ |

## B.4 Advanced Dataset

The advanced dataset was synthetically generated through a prompting process utilizing a larger language model due to the increased complexity of the problems. Specifically, we employed the LLaMA3-70B-Instruct model [18]. The data generation method mirrored that of the basic dataset, with parameters set to a temperature of 1.5, a top_p of 0.9, and a maximum length of 512 tokens, but with variations in the prompts to accommodate the more complex problem types.

Each solution in the advanced dataset follows a structured approach:

1. **Analysis and Summarization**: The system begins by analyzing and summarizing the key points of the word problem.

2. **Problem Type Identification**: It identifies the specific type of mathematical problem presented.

3. **Information Extraction and Reformulation**: Essential variables and information are extracted from the problem, which are then used to reformulate the word problem into a standard form.

4. **Symbolic Representation**: The standard form is converted into a symbolic representation, preparing it for processing by the symbolic solver to derive the solution.

The advanced supervised learning data maintains a structure similar to the basic dataset, consisting of word problems and their conversion to symbolic forms. However, the problems in this dataset are more intricate, including types such as percentage interest problems.

The advanced dataset comprises 79,000 samples, distributed as follows: 60% single-variable and compound inequality problems, 25% two-variable equation problems, and 15% least common multiple and greatest common divisor problems.

**System of equations**

| Question | Answer |
| --- | --- |
| Word Problem: Arnold invested $64,000, some at 5.5% interest and the rest at 9%. How much did he invest at each rate if he received $4,500 in interest in one year? | Break down the word problem into smaller information: 1. Determine how much Arnold invested at each interest rate. 2. Arnold invested $64,000 in total. 3. He invested some at 5.5% interest and the rest at 9%. Define the variables and formulate the system of equations: Let $x$ be the amount invested at 5.5% and $y$ be the amount invested at 9%. The system of equations: $x+y = 64000$ and $0.055*x+0.09*y = 4500$. Symbolic form: $[x + y - 64000, 0.055 * x + 0.09 * y - 4500, x, y, \text{solve}]$ |

**Inequalities**

| Question | Answer |
| --- | --- |
| Word Problem: A taxi charges a flat rate of $1.75, plus an additional $0.65 per mile. If Erica has at most $10 to spend, how far can she travel? | Break down the word problem: 1. Our goal is to find the maximum distance Erica can travel. 2. The taxi charges $1.75 flat and $0.65 per mile. Define the variables and inequality: Let $x$ be the distance traveled. The total cost: $1.75 + 0.65 * x <= 10$. Symbolic form: $[1.75 + 0.65 * x <= 10, x, \text{solve}]$ |

## Compound inequalities

| Question | Answer |
|---|---|
| Word Problem: A ski shop carries skis that are between 150 and 220cm long. They recommend that the skis be 1.25 longer than your height.Calculate the tallest height that a person can be and still rent skis from the shop. | Break down the word problem into smaller information:
1. Our goal is to find the tallest height that a person can be and still rent skis from the shop.
2. The ski shop carries skis that are between 150 and 220cm long.
3. They recommend that the skis be 1.25 longer than your height.
Identify problem type: The ski shop carries skis that are between 150 and 220cm long sets the range of acceptable ski lengths for the customer. This implies the use of compound inequalities to solve the problem.
Define the variables and formulate the compound inequality:
Let x be the height of the person.
between 150 and 220cm long can be written as 150 <= Ski length <= 220
The skis should be 1.25 times longer than your height so the ski length is 1.25*x
Compound Inequality:$150 <= 1.25 * x <= 220$
Symbolic Form:$[[150 <= 1.25 * x, 1.25 * x <= 220, x, solve]]$ |

## Greatest common divisor

| Question | Answer |
|---|---|
| Word Problem: Sara has 16 red flowers and 24 yellow flowers. She wants to make bouquets with the same number of each color. What is the greatest number of bouquets she can make? | Break down the word problem:
1. Find the greatest number of equal bouquets Sara can make.
2. She has 16 red and 24 yellow flowers.
The problem asks for the greatest common divisor of 16 and 24.
Symbolic form: $[16, 24, igcd]$ |

## Least common multiple

| Question | Answer |
|---|---|
| Word Problem: The soccer team plays every three days, and the basketball team plays every five days. When will both teams have games on the same day again? | Break down the word problem:
1. Find when both teams will have games on the same day.
2. The soccer team plays every 3 days, and the basketball team plays every 5 days.
The least common multiple of 3 and 5 is needed.
Symbolic form: $[3, 5, ilcm]$ |

# C Scores Charts

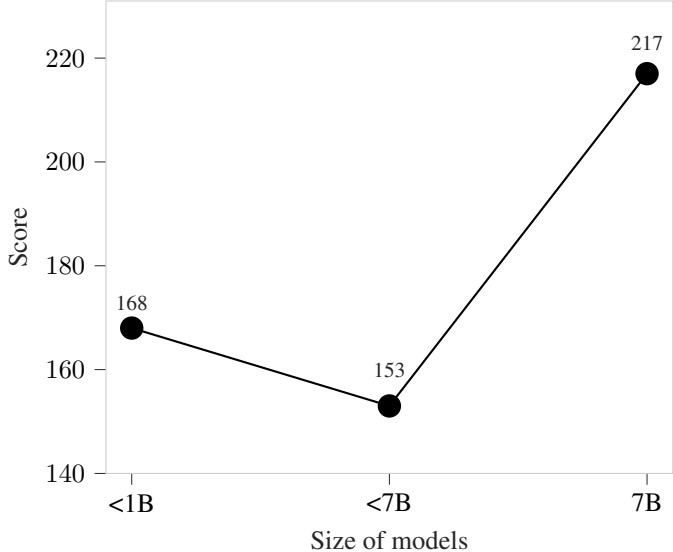

Figure 1: Highest Score of model by size

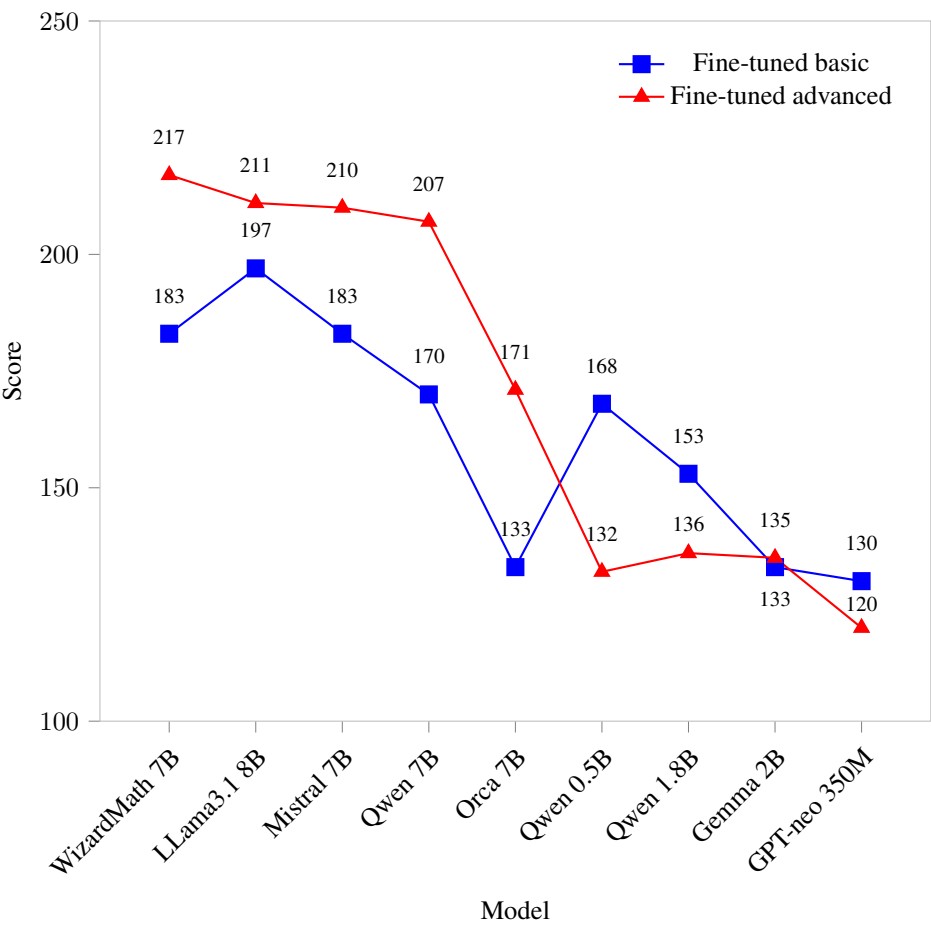

Figure 2: Score of models fine-tuned on basic and advanced datasets

# D  Qwen2Math Comparison of Solutions

Table 4: Comparison of two solutions for finding the greatest possible donation

**Given Problem:** Anthony's mom gave him 30 dollars while Jennifer's mom gave her 36 dollars. They want to give the same amount of money to a charity. What is the greatest donation they can make? To solve this problem, follow the steps provided in the examples.

| Their Solution | Our Solution |
|---|---|
| Step 1:Find the prime factorization of each number.
- 30 = 2 * 3 * 5
- 36 = $2^2$ * $3^2$

Step 2:Identify the common prime factors and their lowest powers.
- The common prime factors are 2 and 3, with the lowest powers being $2^1$ and $3^1$.

The greatest common divisor is the product of the common prime factors.
- GCD = $2^1$ * $3^1$ = 6

Step 3:To find the greatest donation they can make, divide the amount of money each person has by the GCD.
- For Anthony: 30 / 6 = 5
- For Jennifer: 36 / 6 = 6

The greatest donation they can make is the GCD multiplied by the smallest quotient from Step 4.
- Greatest donation = 6 * 5 = 30

**Final Answer:** The greatest donation they can make is 30 dollars. | Break down the word problem into smaller information:
1. Our goal is to find the greatest donation Anthony and Jennifer can make.
2. Anthony's mom gave him 30 dollars.
3. Jennifer's mom gave her 36 dollars.
4. They want to give the same amount of money to a charity.

Identify problem type:
The problem states that Anthony and Jennifer want to give the same amount of money to a charity, which implies that we need to find a number that divides both the amount of money Anthony's mom gave him (30) and the amount of money Jennifer's mom gave her (36) evenly. The problem asks for the greatest donation, indicating that we should use the greatest common divisor to determine the maximum amount they can donate while maintaining equal distribution.

Symbolic Form:
`[[30, 36, igcd]]`

**Final Answer:** 6 |

Table 5: Comparison of two solutions for solving the word problem

| Problem: Eun-Kyung works as a tutor and earns $60 per hour. She has $792 in savings. She is planning an anniversary party for her parents. She would like to invite 40 guests. The party will cost her $1,520 for food and drinks and $150 for the photographer. She will also have a favor for each of the guests, and each favor will cost $7.50. How many hours must she tutor to have enough money for the party? (Assume she will not spend any of her savings.) | |
|---|---|
| **Their Solution** | **Our Solution** |
| Step 1: Calculate the total cost of the party:
Total cost = Cost of food and drinks + Cost of photographer + Cost of favors
Total cost = $1520 + $150 + (40 * $7.50)
Total cost = $1520 + $150 + $300
Total cost = $2070 | Break down the word problem into smaller information:
1. Our goal is to find the number of hours Eun-Kyung must tutor to have enough money for the party.
2. Eun-Kyung earns $60 per hour.
3. She has $792 in savings.
4. The party will cost her $1,520 for food and drinks and $150 for the photographer.
5. She will also have a favor for each of the 40 guests, and each favor will cost $7.50. |
| Step 2: Let x represent the number of hours Eun-Kyung needs to tutor. Write the inequality for the total earnings:
60x >= 2070 | Identify problem type:
The problem asks for the number of hours Eun-Kyung must tutor to have enough money for the party which implies finding the minimum value that satisfies the constraints. The total cost of the party should be greater than or equal to the total funds available is a lower bound constraint. We can use the inequality to solve this problem. |
| Step 3: Solve the inequality:
x >= 2070/60
x >= 34.5

Since Eun-Kyung cannot tutor a fraction of an hour, we round up to the nearest whole number.
x >= 35 | Define the variables and formulate the inequality:
Let x be the number of hours Eun-Kyung must tutor.
Total cost = Cost of food and drinks + Cost of photographer + Cost of favors + Total cost of the party
Total money available = Total funds available + Earnings from tutoring
The party will cost her $1,520 for food and drinks so the cost of food and drinks is 1520
The party will cost her $150 for the photographer so the cost of photographer is 150
She will also have a favor for each of the 40 guests, and each favor will cost $7.50 so the cost of favors is 7.5*40
She has $792 in savings so the total funds available is 792
Eun-Kyung earns $60 per hour so Earnings from tutoring is x*60
Total money available should be greater than or equal to the total cost of the party so the total money available >= 1520 + 150 + 7.5*40
Inequality: 792 + x*60 >= 1520 + 150 + 7.5*40
Symbolic Form: [[792 + x*60, >=, 1520 + 150 + 7.5*40, x, solve]] |
| **Final Answer:** Eun-Kyung must tutor for at least 35 hours to have enough money for the party. | **Final Answer:** x >= 19.63 |

Table 6: Comparison of two solutions for solving the word problem

| | |
|---|---|
| **Problem:** Five student government officers want to go to the state convention. It will cost them $110 for registration, $375 for transportation and food, and $42 per person for the hotel. There is $450 budgeted for the convention in the student government savings account. They can earn the rest of the money they need by having a car wash. If they charge $5 per car, how many cars must they wash in order to have enough money to pay for the trip? | |

| Their Solution | Our Solution |
|---|---|
| Step 1: Calculate the total cost of the trip.
- Registration cost: $110
- Transportation and food cost: $375
- Hotel cost per person: $42 * 5 = $210
- Total hotel cost: $210

Total cost = $110 + $375 + $210 = $795 | Break down the word problem into smaller information:
1. Our goal is to find the number of cars the student government officers must wash to have enough money to pay for the trip.
2. The cost of registration is $110.
3. The cost of transportation and food is $375.
4. The cost of the hotel is $42 per person.
5. There are 5 student government officers.
6. There is $450 budgeted for the convention in the student government savings account.
7. They can earn the rest of the money they need by having a car wash.
8. They charge $5 per car. |
| Step 2: Determine how much more money is needed to meet the budget.
Budgeted amount: $450
Total cost: $795
Additional amount needed = $795 - $450 = $345 | Identify problem type:
The problem asks for the number of cars the student government officers must wash to have enough money to pay for the trip which implies finding the minimum value that satisfies the constraints. The total cost of the trip should be greater than or equal to the total funds available is a lower bound constraint. We can use the inequality to solve this problem. |
| Step 3: Calculate how many cars they need to wash to earn the remaining amount.
Car wash charge per car: $5
Number of cars needed = $345 / $5 = 69 | Define the variables and formulate the inequality:
Let x be the number of cars the student government officers must wash.
Total cost = Cost of registration + Cost of transportation and food + Cost of hotel + Total cost of the trip
Total money available = Total funds available + Earnings from the car wash
The cost of registration is $110 so the cost of registration is 110
The cost of transportation and food is $375 so the cost of transportation and food is 375
The cost of the hotel is $42 per person so the cost of the hotel is 42*5
There is $450 budgeted for the convention in the student government savings account so the total funds available is 450
They can earn the rest of the money they need by having a car wash so Earnings from the car wash is x*5
Total money available should be greater than or equal to the total cost of the trip so the total money available >= 110 + 375 + 42*5
Inequality: 450 + x*5 >= 110 + 375 + 42*5
Symbolic Form: [[450 + x*5, >=, 110 + 375 + 42*5, x, solve]] |
| **Final Answer:** They need to wash 69 cars to have enough money to pay for the trip. | **Final Answer:** x >= 49 |

Table 7: Comparison of two solutions for finding the least number of days to save the same amount

**Given Problem:** Steve can save 9 dollars every day while Maria can save 12 dollars every day. What is the least number of days it will take each person to save the same amount of money? To solve this problem, we need to find the least common multiple (LCM) of the amounts saved by Steve and Maria each day.

| Their Solution | Our Solution |
|---|---|
| 1. Find the LCM of 9 and 12:
- Prime factorization of $9 = 3^2$
- Prime factorization of $12 = 2^2 * 3$
- The LCM is $2^2 * 3^2 = 36$ dollars

2. Determine how many days it will take Steve to save 36 dollars:
- 36 / 9 = 4 days

3. Determine how many days it will take Maria to save 36 dollars:
- 36 / 12 = 3 days

The least number of days it will take each person to save the same amount of money is the LCM of the number of days it takes each to save 36 dollars, which is 12 days.

**Final Answer:** 12 days | Break down the word problem into smaller information:
1. Our goal is to find the least number of days it will take each person to save the same amount of money.
2. Steve can save 9 dollars every day.
3. Maria can save 12 dollars every day.

Identify problem type:
The problem involves two people saving money at different rates. The question asks for the least number of days it will take each person to save the same amount of money, indicating that we need to find the least common multiple of the number of days to determine when they will have the same amount of money.

Symbolic Form:
`[[9, 12, ilcm]]`

**Final Answer:** 36 |

