# OpenReview forum: "Math2Sym: A System for Solving Elementary Problems via Large Language Models and Symbolic Solvers"
_NeurIPS.cc/2024/Workshop/MATH-AI — MATH-AI 24_

### Official Review · Reviewer_mRA7 · 2024-10-05
**Convert MWP solving into symbolic formats**

**Rating:** 7
**Confidence:** 3

**Review:**

The authors propose Math2Sym, a methodology that utilizes sympy to construct symbolic answer forms from MWP for effective solving such questions. Training 7B models over the constructed EMSF dataset show great improvements in solving Math problems with symbolic formats.

Pros:

- Math2Sym successfully integrates the language understanding strengths of LLMs with the precise reasoning abilities of a symbolic solver, i.e., Sympy. And the author further construct the EMSF dataset which enhances model training by providing diverse and structured math problems of varying difficulty.

Cons:

- I understand there has been page limit for submission, but I hope authors could give some more analysis apart from only one score table.
- Constructing EMSF dataset also requires supervision or help from larger models, e.g., (Mixtral 8x7B or Llama-70B); I wonder if authors can make some experiments over this model sizes to check the dataset’s effect.

---

### Official Review · Reviewer_bwqU · 2024-10-06
**A clever hybrid approach**

**Rating:** 7
**Confidence:** 3

**Review:**

### Summary

The paper proposes a hybrid approach for solving math word problems (MWPs) by integrating LLMs with symbolic solvers. The Math2Sym framework enhances MWP-solving by converting natural language descriptions into symbolic form using fine-tuned LLMs, and delegating arithmetic and symbolic manipulations to a SymPy-based solver. They also introduce a new dataset, EMSF, which includes problems across various levels of complexity, and demonstrates that the Math2Sym system significantly improves performance in elementary-level math tasks when compared to other LLM-based approaches.

### Strengths

1. The Math2Sym system effectively combines the strengths of LLMs for language comprehension and symbolic solvers for mathematical precision, resulting in a more accurate and interpretable solution process. This division of tasks allows the LLM to focus on problem interpretation and avoids errors in direct computations.
2. The introduction of the EMSF dataset provides a valuable resource for fine-tuning LLMs to solve MWPs. But some details on the dataset are missing, eg: size of dataset, etc.
3. The paper presents strong empirical results, showing that Math2Sym significantly outperforms baselines such as GPT-3.5 and is competitive with GPT-4-mini on elementary math problems. This improvement is particularly notable in few-shot learning tasks and on more difficult problems.

### Weaknesses

1. The system is primarily evaluated on elementary-level problems, which limits its generalizability to more complex math tasks. Although the paper briefly mentions potential scalability, no experiments are conducted on advanced topics such as calculus or higher-dimensional problems, which would be important for broader adoption.
2. The model is evaluated on synthetic and textbook-inspired datasets, but real-world applications (such as noisy data or problems with ambiguous language) are not explored. This limits the paper’s claims about the system’s robustness in practical settings.
3. The paper discusses performance drops with smaller models on advanced datasets, suggesting that the system’s effectiveness is highly dependent on dataset complexity and model size.

### Questions

1. How does Math2Sym handle ambiguities in problem formulation, such as vague or incomplete descriptions, which are often encountered in real-world problems?
2. Can this system be adapted to solve higher-level mathematical problems beyond the elementary scope, such as calculus or multivariable algebra, and what modifications would be required for this?
3. What are the computational limitations or overheads when using symbolic solvers like SymPy at larger scales? Could this approach still perform efficiently with more complex problems or larger datasets?
4. How robust is Math2Sym in handling noisy or poorly structured input data, which may deviate from textbook-style problems?

---

### Official Review · Reviewer_vKku · 2024-10-08
**Review of Math2Sym**

**Rating:** 4
**Confidence:** 4

**Review:**

Needs improvement:

The idea of using LLM to convert math problems into symbolic forms is not new. Although it is challenging to autoformalize abstract proof based math problems into Isabelle/HOL, the task of converting elementary school word problems into equations which is then delegated to SymPy to be solved is not a challenging research problem but rather one of data curation and careful engineering. Since the class of math problems solvable by the work presented is very limited and elementary, the work lacks major innovation in mathematical reasoning with AI.

---

### Official Review · Reviewer_o8h8 · 2024-10-08
**The paper presents a novel approach combining LLMs with symbolic solvers for math word problems, showing promising results and contributing a new dataset, but lacks clarity in some evaluations and has potential limitations in handling integer-only solutions.**

**Rating:** 7
**Confidence:** 4

**Review:**

The paper presents an interesting and novel approach to solving math word problems by combining LLMs with symbolic solvers. The method shows promising results and has the potential for educational applications. The authors also contribute to a novel EMSF dataset for converting elementary math word problems into symbolic form. At the same time, this paper has some limitations in methodology and evaluation listed below.

# Pros

## Methodology

1. The integration of LLMs with symbolic solvers addresses both linguistic comprehension and precise computation in MWP solving.

2. This approach can solve various types of math problems, beyond simple addition and multiplication problems.

3. The authors contributed to a novel and comprehensive dataset that is proven to be useful in their evaluation chapter.

## Evaluation

1. The authors did extensive evaluation with different settings on different models and the results are sound and reasonable. The results are presented mostly clearly and concisely in table 1.

2. The authors included a few pairs of examples between their approach and Qwen2Math model in the appendix, enabling intuitive understanding of their approach.

## Writings

1. The paper is well-structured and free of grammar mistakes.

# Cons

## Methodology

1. The authors' solver is built using SymPy, but Sympy doesn't directly support specifying integer-only solutions. The authors' test set clearly contains problems that require an integral solution, but it's unclear whether and how the authors dealt with it. Table 6, for example, asks for "how many cars must they wash", implying the answer should be rounded up to the next integer. The symbolic form derived by the authors' approach doesn't specify such constraint. Even though the final answer **happens** to be an integer, it is unclear how this approach could be generalized if symbolic execution gives a floating number as result.


## Evaluation

1. The authors didn't evaluate their approach on larger open-source LLMs. The authors also didn't compare their results with more advanced closed-source LLMs like gpt-4o (although they compared with gpt-4o mini).

2. Table 1 presented "max score" but didn't explain where this score came from, as this number does not appear in elsewhere in the paper. Furthermore, L122-127 mentioned Qwen2-Math-Instruct 7B whose score is not listed except its score in high-difficulty problems.

3. L285, table 4 compares "their solution" (i.e. Qwen2Math) and "our solution" using an example, but a) it's unclear what our solution is - which setting in table 1 does it correspond to? b) the authors simply listed the solutions by both approaches without comments. Table 4 shows an example where "our solution" performs worse than the baseline by giving an incorrect result, but the authors didn't acknowledge, comment on or explain it.

## Writings

1. L361 claims the dataset and code are provided, but I couldn't find either. Since the checklist is not required anyway, I would consider it as an honest mistake and presume the authors would open-source their dataset and code upon acceptance.

2. L28 wrongly refers to GPT-3.5 as an open-source LLM.

---

### Decision · Program_Chairs · 2024-10-09

Accept